# Visual and Ultrasensitive Detection of a Coronavirus Using a Gold Nanorod Probe under Dark Field

**DOI:** 10.3390/bios12121146

**Published:** 2022-12-08

**Authors:** Xuejia Qian, Yuanzhao Shen, Jiasheng Yuan, Chih-Tsung Yang, Xin Zhou

**Affiliations:** 1College of Veterinary Medicine, Institute of Comparative Medicine, Yangzhou University, Yangzhou 225009, China; 2Jiangsu Co-Innovation Center for Prevention and Control of Important Animal Infectious Diseases and Zoonoses, Yangzhou University, Yangzhou 225009, China; 3Joint International Research Laboratory of Agriculture and Agri-Product Safety, The Ministry of Education of China, Yangzhou University, Yangzhou 225009, China; 4Future Industries Institute, Mawson Lakes Campas, University of South Australia, Adelaide, SW 5095, Australia

**Keywords:** porcine epidemic diarrhea virus (PEDV), gold-nanorod probe (GNR probe), dark field microscopy (DFM), low detection limit, on-site detection

## Abstract

Porcine epidemic diarrhea virus (PEDV), a coronavirus that causes highly infectious intestinal diarrhea in piglets, has led to severe economic losses worldwide. Rapid diagnosis and timely supervision are significant in the prophylaxis of PEDV. Herein, we proposed a gold-nanorod (GNR) probe-assisted counting method using dark field microscopy (DFM). The antibody-functionalized silicon chips were prepared to capture PEDV to form sandwich structures with GNR probes for imaging under DFM. Results show that our DFM-based assay for PEDV has a sensitivity of 23.80 copies/μL for simulated real samples, which is very close to that of qPCR in this study. This method of GNR probes combined with DFM for quantitative detection of PEDV not only has strong specificity, good repeatability, and a low detection limit, but it also can be implemented for rapid on-site detection of the pathogens.

## 1. Introduction

Porcine epidemic diarrhea virus (PEDV), a member of the Alphacoronavirus genus from the coronaviradae family, is one of the main causes of highly infectious intestinal diarrhea in swine. PEDV infection results in the high mortality of piglets [1,2]. PEDV has led to devastating damage of the swine industry globally [3]. Rapid detection technologies have been implemented to prevent further spread of PEDV [4]. Currently, fluorescence quantitative PCR (qPCR) remains the gold standard for the diagnosis of PEDV infections [5,6]. Although qPCR provides sensitive, specific, and rapid detection of the viral RNAs in clinical samples, it is tedious to perform RNA extraction and reverse transcription of samples prior to PCR. In addition, a clean environment is needed to prevent sample contamination, and the qPCR instrument is expensive and usually not available in poor areas. All these factors restrict the practical use of qPCR for on-site detection of PEDV.

In recent years, due to the unique properties of nanoparticles, they have been widely employed in various biological systems [7,8]. Metallic nanoparticles (MNPs) are one of the most widely used nanomaterials. Among them, gold nanoparticles (GNPs) are widely used due to their unique optical properties, i.e., localized surface plasmon resonance (LSPR) [9]. The strong light scattering of GNPs at the LSPR frequency makes them very promising for optical imaging and labeling in biological systems [10]. In addition, due to their stability, facile preparation, and easy modification, GNPs are widely used to develop novel detection methods, such as the GNPs-based ELISA assay [11] and the GNP-based DLS analytical method [12]. Recently, dark field microscopy (DFM) combined with GNP probes has been used for the detection of multiple organisms using the rapid readout for color analysis [13,14]. Most imaging techniques require a sophisticated optical setup, such as a laser, optical components, detectors, and complex image processing units [15,16]. However, GNPs-based dark field imaging only requires a dark field concentrator implemented on a common optical microscope, which can significantly reduce the cost. Moreover, it is simple to operate and can be applied for on-site detection of pathogens.

We have previously shown many DFM counting strategies for pathogens of various sizes, such as Cryptosporidium [17], Chlamydia pneumoniae [18], and white spot syndrome virus [19]. These counting strategies make the best use of the dependence of LSPR on the proximity of other nanoparticles. Small nanoparticles (<30 nm) that cannot be observed clearly under a dark field clustered to form bright wreath-like structure due to the presence of bio-analytes. Yet, GNPs larger than 40 nm in diameter can be easily observed with the naked eye using a dark field (optical scattering) microscope due to the high scattering cross-section of large GNPs, and this highly enhanced cross-section provides sensitive and highly contrasting images [20,21].

The amount of GNPs is related to the content of the target through specific reactions, which enables rapid, low-cost, highly sensitive, and visible detection of targets, such as DNA [22], RNA [23], miRNAs [24], and proteins [25]. Herein, we propose a gold nanorod probe (GNR probe)-assisted antigen-counting strategy to quantify PEDV. The fluorophores used in qPCR are susceptible to quenching, while plasmonic nanoparticles do not flicker or bleach, providing a nearly infinite photon budget for observing molecular binding at long intervals [26]. Notably, owing to its increase in longitudinal LSPR, GNRs (aspect ratio of 3) are six times more sensitive than nanospheres [27]. Accordingly, GNRs are used in this study.

Our strategy includes three steps, as illustrated in Figure 1: (1) functionalized GNR probes; (2) antibody-modified silicon chips prepared by standard chemical modification; and (3) DFM for counting.

## 2. Materials and Methods

### 2.1. Chemicals and Instruments

PEDV CV777 strain was stored in our lab. Protein G-conjugated gold nanorods (120 nm in length and 40 nm in width, ~1.0 × 10^12^ particles/mL) were purchased from Creative Diagnostics. (New York, USA). Mouse anti-PEDV polyclonal antibodies were prepared as described in Section 2.2. BSA was purchased from Sigma-Aldrich (St. Louis, MO, USA). PBS was supplied by Biyuntian Co. (Shanghai, China). Tween 20 was received from Aladdin (Los Angeles, CA, USA).

Transmission electron microscopy (TEM) images were obtained using a Tecnai 12 transmission electron microscope (Philips, AMS, The Netherlands). Dark field images were obtained using a Nikon DFM (Nikon, Tokyo, Japan). SEM images were obtained using a GeminiSEM 300 (Carl Zeiss, Oberkochen, Germany) with an acceleration voltage of 10.0 kV and 5 k of magnification. The zeta potential was measured using a Malvern Instrument (Zetasizer NanoES90, Worcs, UK). All fluorescence quantitative data were measured by a LightCycler480 II fluorescence quantitative PCR (qPCR) instrument (Roche, Basel, Switzerland).

### 2.2. Preparation of Anti-PEDV Polyclonal Antibody

The strategy for the preparation of anti-PEDV polyclonal antibody was based on the literature with slight modification [28]. The construction of a pET-His-S1 plasmid was used to express the antigen protein for immunization. Briefly, the gene S1 of PEDV was amplified by primers (Forward primer: 5′-TGACAAGCTTACTAACTTTAGGCGGTTCTT-3′; Reverse primer: 5′-TGACATGGAACATAGCCAATACTGC-3′) and inserted into pET-28a (+). The pET-His-S1 clones were transformed into *E. coli* BL21 cells by heat excitation. The individual colony was picked from the LB plate, which contained 100 μg/mL of kanamycin solution, followed by the identification of the sequence of the pET-His-S1 plasmid based on colony PCR and further DNA sequencing. Subsequently, it was transferred to 5 mL of LB medium for cultivation overnight on a shaking incubator at 37 °C and 250 rpm. On the next day, 0.25 mM Isopropyl-β-D-thiogalactopyranoside (IPTG) was used to induce pET-His-S1 plasmid when the cell density reached the absorbance of 0.6–0.8 at OD 600 nm. The cell pellets were collected by the centrifugation of the induced cell culture suspension at 5000 g for 5 min, followed by resuspension in PBS. To further retrieve the protein, the standard ice bath ultrasonication was performed to crush cells, and the solution was centrifuged at 12,000× *g* rpm for 10 min. Finally, the protein was purified by Ni^2+^-NTA resin through the resuspension of precipitation in urea solution, followed by renaturation. The antibody was produced by mixing the purified protein with Freund’s adjuvant for the immunization of BALB/c mouses.

### 2.3. Preparation of the Anti-PEDV Antibodies Modified Capture Chip

The preparation of anti-PEDV antibodies modified capture chip was based on the literature [29]. In brief, the silicon chips (0.3 cm^2^) were immersed in a freshly made piranha solution (volume ratio of 30% H_2_O_2_ to 18M H_2_SO_4_ is 3:7) for 1 h. (*Caution: the preparation of piranha solution is exothermic and please follow the standard operation procedure*). Afterward, they were washed 6 times with DI water and immediately immersed in a solution containing 15 mL of anhydrous ethanol and 1 mL of APTES for 2 h at 37 °C for surface functionalization. The immobilized primary amines of APTES were then activated with 10% of glutaraldehyde for 1 h, followed by immersion in 15 μg/mL of PEDV antibody solution at 37 °C for 4 h. Finally, the chips were blocked with BSA (2 mg/mL) for 1 h and rinsed with PBST containing 5% of Tween 20 in PBS for 3 times to obtain anti-PEDV-immobilized chips. The antibody-functionalized chips were stored at 4 °C prior to use.

### 2.4. Preparation of Specific GNR Probes

It is well known that Protein G can bind specifically to the Fc segment of antibody. The GNR probes were prepared by mixing 100 μL of protein G-conjugated GNRs with 10 μL of PEDV antibody. It is critical to optimize the ratio of GNR and antibody on the probe to compromise the binding efficiency and stability of the probe. To determine the optimal concentration ratio of GNR to antibody, 5 different concentrations (1, 3, 6, 9, and 12 μg/μL) of antibody were selected to interact with GNRs. After incubation for 4 h at room temperature, the functionalized GNR probes were washed with PBS for 3 times and centrifuged at 6000× *g* for 15 min to remove unattached anti-PEDV antibodies. The successful modification of antibodies on the probe was measured by a Malvern Zetasizer and SDS-PAGE.

### 2.5. Sandwich Immunoassay on the Capture Chip

10 μL of PEDV at various concentrations (3.07 × 10^1^, 1.53 × 10^2^, 7.67 × 10^2^, 3.83 × 10^3^, and 1.92 × 10^4^ copies/μL in PBS buffer) or real samples were dropped on the capture chips. The chips were placed in a 96-well plate and incubated at 37 °C for 30 min. The chips were thoroughly washed with PBST for 5 times. Subsequently, GNR probes (100 μL, 1 nM) were incubated with the anti-PEDV chips to capture PEDV particles to form the sandwich immunoassay at 37 °C for 10 min. Before DFM counting, PBS was used to wash off unattached GNR probes from the chips, and ammonium acetate solution was used to remove sodium salts in PBS to reduce the background signal in DFM.

### 2.6. GNR Probe-Assisted DFM Counting Strategy for Detection of PEDV Samples

Through the GNR probe-assisted DFM counting strategy, the relative ratio of the number of green halos versus the concentration of PEDV can be measured quantitatively. The number of green halos should increase accordingly with the increased amount of antigen in the sample solution, and this correlation will lay the analytical foundation of a DFM counting strategy. Notably, 10 μL of sample covers the entire area of the chip (9 mm^2^), which is about 400 times the size of a single field of view (0.0225 mm^2^). The number of green halos from 20 random field of views on each chip was averaged. Then, we can determine the number of green halos in a 10 μL of PEDV sample (namely, 400 fields of view). Finally, the virus concentration (copies per microliter) can be calculated from the standard curve of the number of green halos on a chip in relation to the virus concentration.

To accurately quantify the number of GNRs, we developed a counting software for accurately identifying and counting green halos formed by GNR in dark field images (Figure 2A). In addition, the image processing software can remove impurities that look similar but are different in color and size from the green halos formed by GNR. First, we investigated the RGB value of GNR luminescence under the DFM and found that the RGB value ranges from (20, 20, 20) to (77, 255, 255). We extracted GNR features such as color, size, and shape by analyzing the connected domain within this value range, and converted the image into grayscale to highlight the outline of the target, namely GNR. Then, after binarization, the image was expanded and corroded several times to eliminate the noise in the image as much as possible. At this time, the target suspected to be GNR was found through the area of the connected domain (greater than 45 pixels and less than 800 pixels). If the RGB value of the center point of the suspected target is between (0, 0, 0) and (20, 20, 20), the target is identified as a hollow ring of green halo generated by a GNR. For example, the RGB value of the center of the circle is green with a RGB value of (68, 201, 75) (Figure 2B), which is obviously out of the RGB range of expected GNR. Thus, it is determined a non-target. The RGB value of the center point of the green circle is black with a RGB value of (10, 8, 4) (Figure 2C); thus, it is considered a true halo generated by a GNR. In addition, Appendix A shows three representative images: an original dark field image, the image sorted by our counting software, and a local enlargement in the counting image.

### 2.7. Sensitivity of GNR Probe-Assisted DFM Counting Strategy

Samples were prepared by diluting the purified PEDV into a series of concentrations at a 5-fold gradient. These samples were used to determine the limit of detection (LOD) of the GNR probe-assisted DFM counting strategy. According to our previously work [30], the LOD of the DFM counting method was determined to be the sample concentration corresponding to a signal-to-noise ratio greater than or equal to 3.

### 2.8. Preparation of Simulated Real Samples for DFM Counting

To validate the feasibility of our DFM counting method for real samples, 5 simulated virus samples with different theoretical concentrations (1.92 × 10^4^ copies/μL, 1.92 × 10^3^ copies/μL, 1.92 × 10^2^ copies/μL, 9.6 × 10^1^ copies/μL, and 2.4 × 10^1^ copies/μL) were obtained by mixing pure PEDV samples with SPF mouse serum. The LOD of our DFM counting method for these spiked samples was compared with that of PCR detection at the same time.

## 3. Results and Discussion

### 3.1. Specificity of Anti-PEDV Polyclonal Antibody

The specificity of the self-made antibody was determined by indirect fluorescent assay (IFA). The self-made anti-PEDV antibody that could be conjugated with goat anti-mouse IgG (Alexa Fluor^®^ 647) was used as the primary antibody, and the serum of unimmunized mice was used as the negative control. As shown in Figure 3, only the PEDV-positive and antibody-positive group (Figure 3A–C) shows obvious red fluorescence, while neither PEDV-negative and antibody-positive group (Figure 3D–F) nor PEDV-positive and antibody-negative group (Figure 3G–I) show obvious red fluorescence. This suggests that the self-made mouse anti-PEDV antibody could react specifically with PEDV.

### 3.2. Characterization and Optimization of GNR Probes

To demonstrate the successful conjugation of anti-PEDV antibodies on nanoparticles, the hydrodynamic dimensions and zeta potentials of the probes were characterized using a Malvern Zetasizer. As illustrated in Figure 4A, the hydrodynamic size distribution shows that the size of the GNR probes was larger than GNR without antibody modification, which is in good agreement with the principle that surface modification of nanoparticles will increase the hydrodynamic size. In addition, with the same particle number, the zeta potential (Figure 4B) of probes was obviously higher than that of the unmodified GNRs due to the negative nature of the antibody.

To optimize the binding ratio of antibodies to GNR probes and determine the binding efficiency of antibodies to GNRs, SDS-PAGE gel analysis (Figure 4C) was performed to measure the optimal antibody concentration. Electrophoresis results show the presence of 2 bands (a ~55 kDa band of antibody heavy chain and a ~20 kDa band of antibody light chain) in the GNR probes, and the optimal antibody concentration is 6 μg/μL, confirming the successful conjugation of the antibody and GNR. Regarding the stability of the GNR probe, TEM images show that the monochromatic dispersion of GNR probe (Figure 4D) is comparable to that of GNR (Figure 4E).

### 3.3. Specificity of GNR Probes

To demonstrate that our GNR probe-assisted counting strategy can be used to detect target viruses in samples, various virus particles—including PEDV, PRRSV (porcine reproductive and respiratory syndrome virus), H9N2 (a subtype of avian influenza virus), and NDV (Newcastle disease virus)—were used to investigate the specificity of the probe. As shown in the TEM images, there was specific binding of probes to PEDV particles (Figure 5A), but the probes were unable to recognize the other three virus particles (Figure 5B–D).

### 3.4. Characterization of the Capture Chips

We next investigated the feasibility of the GNR probe-assisted counting strategy with SEM. Compared with the antibody-modified chip without PEDV incubation as a control (Figure 6A), the chip incubated with PEDV could be labeled by GNRs, as shown in SEM imaging (Figure 6B). The images proved that the chips can capture PEDV and form sandwich structures.

### 3.5. GNR Probe-Assisted DFM Counting of PEDV in PBS

We first verified the feasibility of the probe to bind with PEDV particles in PBS under TEM, and then verified the feasibility of the chip-based sandwich immunoassay using SEM. Prior to this, we verified that GNRs on silicon wafer surfaces presented in the form of a green halo under the dark field imaging (Appendix A). As shown in Figure 7A–F, the number of GNR particles on the silicon chip was significantly correlated with the increased concentrations of PEDV, suggesting the good consistency of the GNR probe-assisted sandwich assay. In addition, we also established a qPCR method (the standard quantification curve was shown in Appendix A and the primers were shown in Appendix A) to detect PEDV samples. Of note, compared to our dark field counting method, qPCR is more time-consuming. Usually, it takes approximate 2.5 h to analyze a sample with qPCR, including the extraction of the genome and reverse transcription (1 h), and the whole PCR program (1.5 h). The counting GNR numbers and qPCR concentrations were plotted as a function of PEDV concentration, respectively, in Figure 7G. The GNR numbers were calculated by multiplying the mean number of green halos in 20 randomly selected DFM field of views from 3 independent experiments by 400 (the surface area of each chip is approximately up to 400 times that of a single field of view). Results (Figure 7G) show the same trend for these two methods. LODs of these two methods are determined by their calibration curves. The theoretical LOD of the GNR probe-assisted DFM counting method is determined by extrapolating the linear curve corresponding to three times of blank noise. The sample without PEDV is used as a control to determine the noise level. The GNRs value averaged from 3 independent experiments is 2 of green halos in 20 fields of view. Therefore, the noise is determined to be 120 of GNR particles. Consequently, the LOD is 22.2 copies/µL according to the equation Y = 10.4X − 111.0 (Figure 7G). Meanwhile, the LOD by PCR for the detection of PEDV is the virus concentration value corresponding to the CT value of 35, which can determine the copy number of the virus gene according to the concentration-CT linear equation (Appendix A, Y = −3.408X + 38.53). The calculated LOD value is 14.4 copies/µL based on the equation of Y = 1.0545X − 4.3 (Figure 7G). Therefore, the GNR probe-assisted DFM counting method shows comparable sensing performance as qPCR, allowing for the detection of virus at low concentrations.

### 3.6. GNR Probe-Assisted DFM Counting of PEDV in Simulated Real Samples

Simulated samples containing different concentrations of PEDV in the complex biological matrix were prepared to validate our GNR probe-assisted DFM counting strategy. Various concentrations of PEDV used in the dark field counting method were also determined by qPCR to compare the consistency and sensing performance of these two methods (Figure 8A). The calculated LOD value of two methods in simulated real samples is 23.80 and 15.5 copies/µL, based on the equation of Figure 8A. According to the histogram statistics (Figure 8B), the data obtained based on our counting method are highly consistent with those from the qPCR method. Taken together, our GNR probe-assisted DFM counting method has good feasibility, validity, and applicability for samples containing a low concentration of virus in the complex biological matrix.

## 4. Conclusions

Our study demonstrated a reliable, rapid, and low-cost GNR probe-assisted dark field counting strategy for quantification of PEDV with a limit of detection (LOD) of 23.80 copies/μL for simulated real samples, which is comparable to the sensitivity (LOD of 15.53 copies/μL) of qPCR in this work. The results of the GNR probe-assisted dark field counting strategy were reliable and highly consistent with the results acquired by qPCR, which demonstrated that the method could be applied for practical use in the clinic. In addition, our counting strategy can exclude the preprocessing of RNA virus detection in qPCR, that is, the extraction of viral RNA and reverse transcription, in turn making our method applicable to on-site detection without any sophisticated process and technicians. For the detection of a single sample, the time cost of our GNR probe-assisted DFM count method is 1h, which is less than that (2.5 h) of qPCR. Taken together, the proposed GNR probe-assisted dark field counting chip platform has the potential to be used as a general tool for pathogen quantification in the field.

## Figures and Tables

**Figure 1 biosensors-12-01146-f001:**
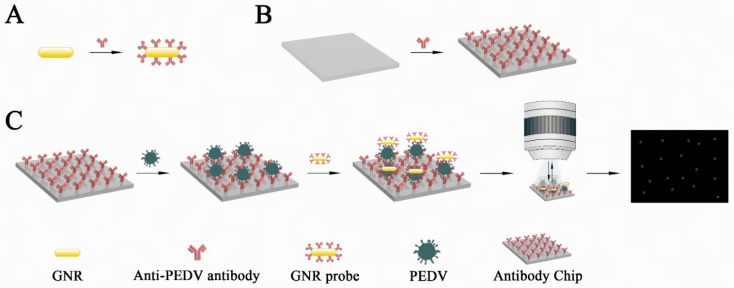
Scheme of GNR probe-assisted DFM counting chip. (**A**) Preparation of GNR probes. (**B**) Functionalization of the chip with antibodies for capturing PEDV. (**C**) Procedures for counting virus particles using GNR probe-assisted DFM counting chip. The capture chip was first functionalized with the antibody through chemical modification, followed by blocking with bovine serum albumin (BSA). In the presence of target viruses, the antibodies on the chip recognized the viruses to form sandwich structures with GNR probes. Each GNR probe of sandwiches presents green halo under DFM. The number of GNR probes can be counted readily.

**Figure 2 biosensors-12-01146-f002:**
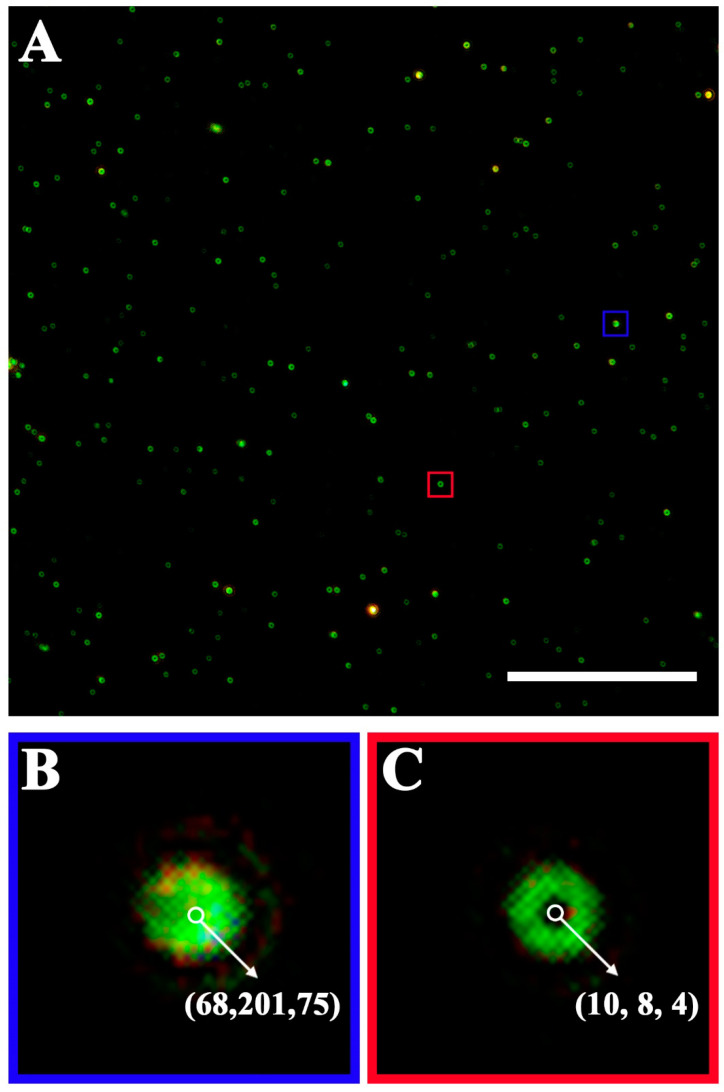
Identification of green halo generated by GNR. (**A**) A DFM image of the captured PEDVs labeling with GNR probes assayed by our counting software. (**B**) The enlargement of the selected area in the blue square of (**A**) and the RGB value of the center point of the green point is (68, 201, 75) by our counting software. (**C**) The enlargement of the selected area in the red square of (**A**) and the RGB value of the center point of the green halo is (10, 8, 4) by our counting software. Scale bar: 20 μm.

**Figure 3 biosensors-12-01146-f003:**
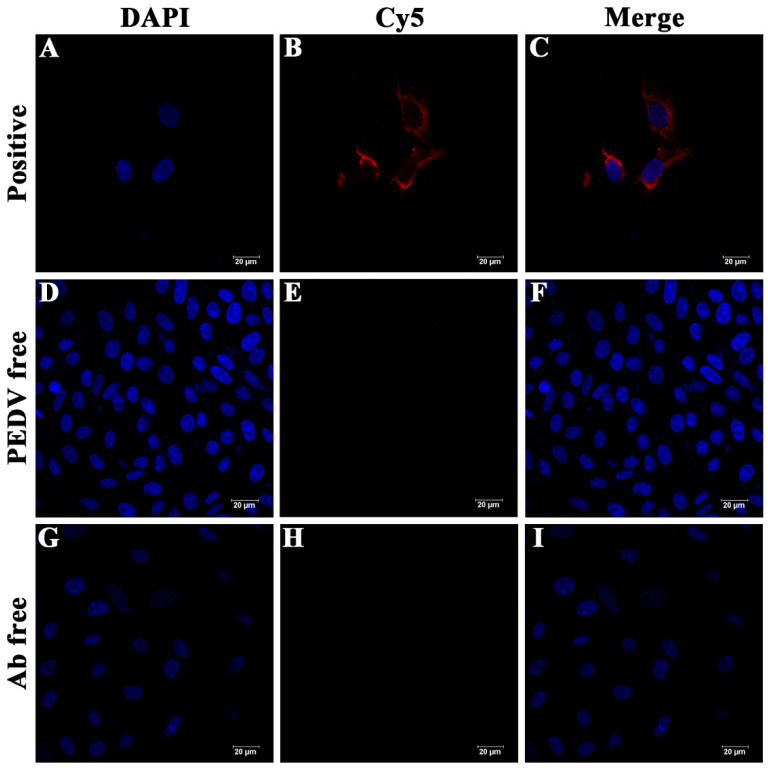
Cell immunofluorescence analysis of anti-PEDV antibody. (**A**–**C**): PEDV-positive and antibody-positive group: PEDV-infected Vero cells were incubated with anti-PEDV antibodies; (**D**–**F**): PEDV-negative and antibody-positive group: uninfected Vero cells were incubated with anti-PEDV antibodies; (**G**–**I**): PEDV-positive and antibody-negative group: PEDV-infected Vero cells were incubated with negative serum of unimmunized mice. Subsequently, all three groups were incubated with goat anti-mouse IgG (Alexa Fluor^®^ 647). Microscopic images showing that goat anti-mouse IgG (Alexa Fluor^®^ 647) only had reactions with PEDV-positive and antibody-positive group. Blue: DAPI-stained DNA; red: Goat anti-mouse IgG.

**Figure 4 biosensors-12-01146-f004:**
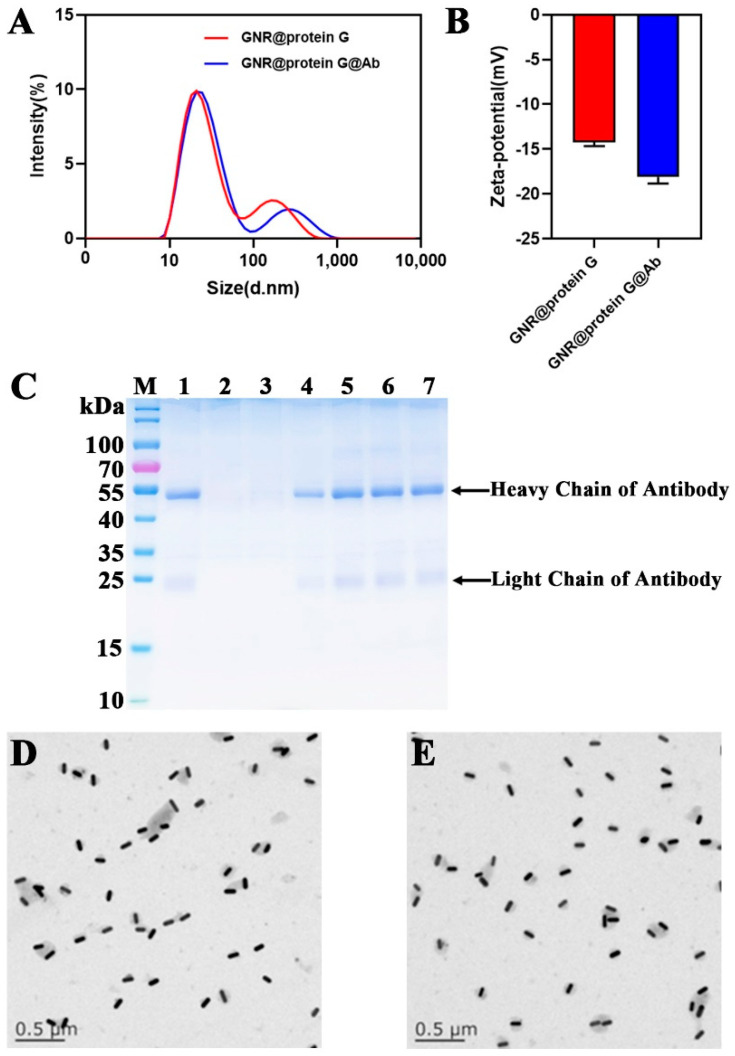
Characterization of GNR probes. (**A**) Hydrodynamic sizes of GNR@protein G and GNR@protein G@Ab. (**B**) Zeta potentials of GNR@protein G and GNR@protein G@Ab. (**C**) SDS-PAGE assay: M, marker; 1, 5.0 μg antibodies; 2, GNR; 3, 1 μg/μL; 4, 3 μg/μL; 5, 6 μg/μL; 6, 9 μg/μL; 7, 12 μg/μL. (**D**) TEM image of GNR probes. (**E**) TEM image of GNRs.

**Figure 5 biosensors-12-01146-f005:**
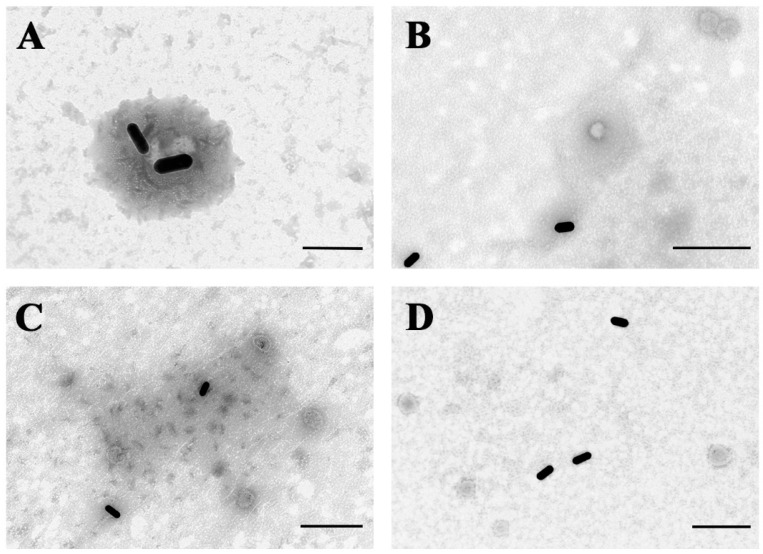
Validation of the specificity of GNR probes based on TEM images. (**A**) GNR probes bound PEDV. (**B**) GNR probes mixed with PRRSV. (**C**) GNR probes mixed with H9N2. (**D**) GNR probes mixed with NDV. Scale bar: 200 nm (**A**) and 500 nm (**B**–**D**).

**Figure 6 biosensors-12-01146-f006:**
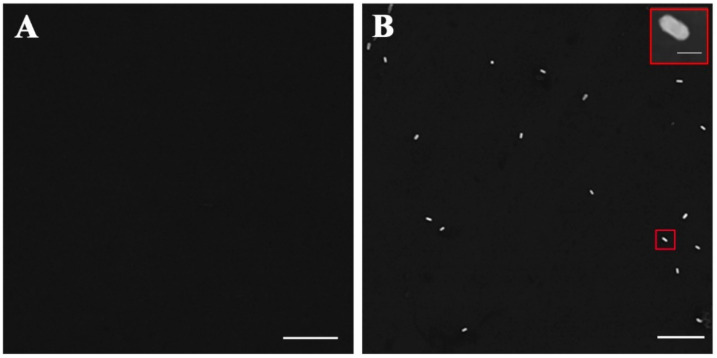
SEM characterization of antibody-modified chips for the capture of PEDV. (**A**) SEM image of antibody-modified chip incubated with GNR probes without PEDV. (**B**) SEM image of the captured PEDV labeling with GNR probes. The upper right corner of the image shows the enlargement of the selected area. Scale bar: 1 μm (**A**,**B**) and 100 nm (the upper right corner in (**B**)).

**Figure 7 biosensors-12-01146-f007:**
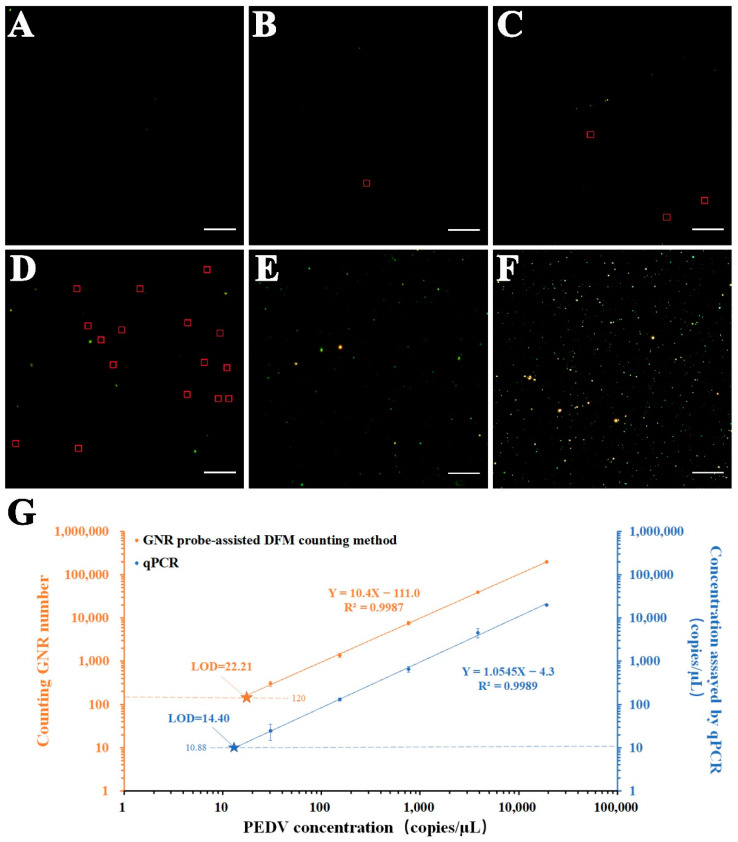
GNR probe-assisted DFM counting of PEDV samples. (**A**) Dark field image of sample without PEDV. (**B**) Dark field image of sample with PEDV of 3.07 × 10^1^ copies/μL. (**C**) Dark field image of sample with PEDV of 1.53 × 10^2^ copies/μL. (**D**) Dark field image of sample with PEDV of 7.67 × 10^2^ copies/μL. The red squares are the green halos generated by a GNR recognized by counting software in B-D. (**E**) Dark field image of sample with PEDV of 3.83 × 10^3^ copies/μL. (**F**) Dark field image of sample with PEDV of 1.92 × 10^4^ copies/μL. (**G**) Calibration curves of DFM counting and qPCR. Each counting GNR number on the GNR probe-assisted counting method curve was obtained by multiplying the mean number of GNR particles from 20 DFM images of the corresponding PEDV sample on the chip by 400. The calibration curve of qPCR was plotted by the quantitative value (theoretical concentration) of RNA extracted from these PEDV samples versus the measured concentration values by the standard curve of the copy number corresponding to the concentration of the standard plasmid-CV777 (a specific gene fragment of PEDV). The pentagram is the intersection of the calibration curve with three times signal-to-noise ratio. Scale bar: 20 μm.

**Figure 8 biosensors-12-01146-f008:**
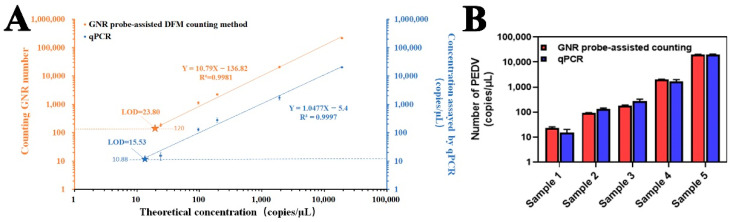
GNR probe-assisted counting of PEDV in simulated real samples. Representative DFM image of PEDV antibody-immobilized chip incubated with GNR probe at different concentrations of PEDV present in mouse serum. (**A**) Calibration curves of DFM counting and qPCR. The pentagram is the intersection of the calibration curve with three times signal-to-noise ratio. (**B**) Comparison of the results of GNR probe-assisted counting method with those of qPCR method. The concentrations of sample 1–5 are 2.4 × 10^1^ copies/μL, 9.6 × 10^1^ copies/μL, 1.92 × 10^2^ copies/μL, 1.92 × 10^3^ copies/μL and 1.92 × 10^4^ copies/μL, respectively. The values of GNR probe-assisted counting method are calculated from the standard curve in Figure 7G.

## Data Availability

The data that support the findings of this study are available from the corresponding author upon reasonable request.

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
