# Peer review of "Visual and Ultrasensitive Detection of a Coronavirus Using a Gold Nanorod Probe under Dark Field"

_biosensors, 2022, doi:10.3390/bios12121146_

Round 1

Reviewer 1 Report

1.      It is better to compare the detection result of DFM and SEM at certain concentration of PEDV, to confirm the reliability of DFM method.

2.      The size of GNRs directly decide the its color in dark field observation. Since the conjugation of antibody and GNRs will arise the aggregration of GNRs, this will give different signal of DMF detection, this should be taken into consideration.

3.      This methods need take some time count the color dots, and strongly interfered with the background signa, the advantage of this method?

Reviewer 2 Report

Authors demonstrated counting of corona virus using antibody attached gold nanorods. I found this paper very useful and suggest to publish with a minor corrections:

1. There is a lack of references in Introduction. Please add appropriate references.

2.Figure 4E is missing the caption.

3. English grammar needs to be improved.

4. In Figure 6A and 6B, SEM images are too dark to see the background. I suggest to coat the surface with Al, Au or Pt to increase the conductivity for a better images.

Round 2

Reviewer 1 Report

this revised manuscript can be accepted now.